# DYNAMIC NEURAL PROGRAM EMBEDDINGS FOR PROGRAM REPAIR

**Ke Wang**[*]
University of California
Davis, CA 95616, USA
`kbwang@ucdavis.edu`

**Rishabh Singh**
Microsoft Research
Redmond, WA 98052, USA
`risin@microsoft.com`

**Zhendong Su**
University of California
Davis, CA 95616, USA
`su@ucdavis.edu`

## ABSTRACT

Neural program embeddings have shown much promise recently for a variety of program analysis tasks, including program synthesis, program repair, code-completion, and fault localization. However, most existing program embeddings are based on syntactic features of programs, such as token sequences or abstract syntax trees. Unlike images and text, a program has well-defined semantics that can be difficult to capture by only considering its syntax (*i.e.* syntactically similar programs can exhibit vastly different run-time behavior), which makes syntax-based program embeddings fundamentally limited. We propose a novel *semantic* program embedding that is learned from program execution traces. Our key insight is that program states expressed as sequential tuples of live variable values not only capture program semantics more precisely, but also offer a more natural fit for *Recurrent Neural Networks* to model. We evaluate different syntactic and semantic program embeddings on the task of classifying the types of errors that students make in their submissions to an introductory programming class and on the CodeHunt education platform. Our evaluation results show that the semantic program embeddings significantly outperform the syntactic program embeddings based on token sequences and abstract syntax trees. In addition, we augment a search-based program repair system with predictions made from our semantic embedding and demonstrate significantly improved search efficiency.

## 1 INTRODUCTION

Recent breakthroughs in deep learning techniques for computer vision and natural language processing have led to a growing interest in their applications in programming languages and software engineering. Several well-explored areas include program classification, similarity detection, program repair, and program synthesis. One of the key steps in using neural networks for such tasks is to design suitable program representations for the networks to exploit. Most existing approaches in the neural program analysis literature have used *syntax-based* program representations. Mou et al. (2016) proposed a convolutional neural network over abstract syntax trees (ASTs) as the program representation to classify programs based on their functionalities and detecting different sorting routines. DeepFix (Gupta et al., 2017), SynFix (Bhatia & Singh, 2016), and sk_p (Pu et al., 2016) are recent neural program repair techniques for correcting errors in student programs for MOOC assignments, and they all represent programs as sequences of tokens. Even program synthesis techniques that generate programs as output, such as RobustFill (Devlin et al., 2017), also adopt a token-based program representation for the output decoder. The only exception is Piech et al. (2015), which introduces a novel perspective of representing programs using input-output pairs. However, such representations are too coarse-grained to accurately capture program properties — programs with the same input-output behavior may have very different syntactic characteristics. Consequently, the embeddings learned from input-output pairs are not precise enough for many program analysis tasks.

Although these pioneering efforts have made significant contributions to bridge the gap between deep learning techniques and program analysis tasks, syntax-based program representations are fundamentally limited due to the enormous gap between program syntax (*i.e.* static expression) and

---

[*]Work done during an internship at Microsoft Research.

```
static int[] BubbleSort(int[] A) {
    int left = 0;
    int right = A.Length - 1;

    for (int i = right;i > left;i--) {
        for (int j = left;j < i;j++) {
            if (A[j] > A[j + 1]) {
                int tmp = A[j];

                A[j] = A[j + 1];
                // instrumentation line
                Console.WriteLine(
                    string.Join(",", A)
                );

                A[j + 1] = tmp;
                // instrumentation line
                Console.WriteLine(
                    string.Join(",", A)
                );
    }}}

    return A;
}
```

```
static int[] InsertionSort(int[] A) {
    int left = 0;
    int right = A.Length;

    for (int i = left;i < right;i++) {
        for (int j = i - 1;j >= left;j--) {
            if (A[j] > A[j + 1]) {
                int tmp = A[j];

                A[j] = A[j + 1];
                // instrumentation line
                Console.WriteLine(
                    string.Join(",", A)
                );

                A[j + 1] = tmp;
                // instrumentation line
                Console.WriteLine(
                    string.Join(",", A)
                );
    }}}

    return A;
}
```

| Bubble | Insertion |
|---|---|
| [5,5,1,4,3] | [5,5,1,4,3] |
| [5,8,1,4,3] | [5,8,1,4,3] |
| [5,1,1,4,3] | [5,1,1,4,3] |
| [5,1,8,4,3] | [5,1,8,4,3] |
| [1,1,8,4,3] | [5,1,4,4,3] |
| [1,5,8,4,3] | [5,1,4,8,3] |
| [1,5,4,4,3] | [5,1,4,3,3] |
| [1,5,4,8,3] | [5,1,4,3,8] |
| [1,4,4,8,3] | [1,1,4,3,8] |
| [1,4,5,8,3] | [1,5,4,3,8] |
| [1,4,5,3,3] | [1,4,4,3,8] |
| [1,4,5,3,8] | [1,4,5,3,8] |
| [1,4,3,3,8] | [1,4,3,3,8] |
| [1,4,3,5,8] | [1,4,3,5,8] |
| [1,3,3,5,8] | [1,3,3,5,8] |
| [1,3,4,5,8] | [1,3,4,5,8] |

Figure 1: Bubble sort and insertion sort (code highlighted in shadow box are the only syntactic differences between the two algorithms). Their execution traces for the input vector $A = [8, 5, 1, 4, 3]$ are displayed on the right, where, for brevity, only values for variable A are shown.

```
1  static int max(int[] arr) {
2
3      int max_val = int.MinValue;
4
5      foreach(int item in arr)
6      {
7          if (item > max_val)
8              max_val = item;
9      }
10
11     return max_val;
12 }
```

| Variable Trace | State Trace |
|---|---|
| $\{max\_val : -\infty\}$ | $\{max\_val : -\infty, item : \bot\}$ |
| $\{item : 1\}$ | $\{max\_val : -\infty, item : 1\}$ |
| $\{max\_val : 1\}$ | $\{max\_val : 1, item : 1\}$ |
| $\{item : 5\}$ | $\{max\_val : 1, item : 5\}$ |
| $\{max\_val : 5\}$ | $\{max\_val : 5, item : 5\}$ |
| $\{item : 3\}$ | $\{max\_val : 5, item : 3\}$ |

Figure 2: Example for illustrating program dependency.

Table 1: Variable and state traces obtained by executing function $max$, given arr = $[1, 5, 3]$.

semantics (*i.e.* dynamic execution). This gap can be illustrated as follows. First, when a program is executed at runtime, its statements are almost never interpreted in the order in which the corresponding token sequence is presented to the deep learning models (the only exception being straightline programs, *i.e.*, ones without any control-flow statements). For example, a conditional statement only executes one branch each time, but its token sequence is expressed sequentially as multiple branches. Similarly, when iterating over a looping structure at runtime, it is unclear in which order any two tokens are executed when considering different loop iterations. Second, program dependency (*i.e.* data and control) is not exploited in token sequences and ASTs despite its essential role in defining program semantics. Figure 2 shows an example using a simple $max$ function. On line 8, the assignment statement means variable $max\_val$ is data-dependent on $item$. In addition, the execution of this statement depends on the evaluation of the $if$ condition on line 7, *i.e.*, $max\_val$ is also control-dependent on $item$ as well as itself. Third, from a pure program analysis standpoint, the gap between program syntax and semantics is manifested in that similar program syntax may lead to vastly different program semantics. For example, consider the two sorting functions shown in Figure 1. Both functions sort the array via two nested loops, compare the current element to its successor, and swap them if the order is incorrect. However, the two functions implement different algorithms, namely *Bubble Sort* and *Insertion Sort*. Therefore minor syntactic discrepancies can lead to significant semantic differences. This intrinsic weakness will be inherited by any deep learning technique that adopts a syntax-based program representation.

To tackle this aforementioned fundamental challenge, this paper proposes a novel *semantic* program embedding that is learned from the program's runtime behavior, *i.e.* dynamic program execution traces. We execute a program on a set of test cases and monitor/record the program states comprising of variable valuations. We introduce three approaches to embed these dynamic executions: (1) *variable trace embedding* — consider each variable independently, (2) *state trace embedding* — consider sequences of program states, each of which comprises of a set of variable values, and (3) *hybrid embedding* — incorporate dependencies into individual variable sequences to avoid redundant variable values in program states.

Our novel program embeddings address the aforementioned issues with the syntactic program representations. The dynamic program execution traces precisely illustrate the program behaves at runtime, and the values for each variable at each program point precisely models the program semantics. Regarding program dependencies, the dynamic execution traces, expressed as a sequential list of tuples (each of which represents the value of a variable at a certain program point), provides an opportunity for Recurrent Neural Network (RNN) to establish the data dependency and control dependency in the program. By monitoring particular value patterns between interacting variables, the RNN is able to model their relationship, leading to more precise semantic representations.

Reed & De Freitas (2015) recently proposed using program traces (as a sequence of actions/statements) for training a neural network to learn to execute an algorithm such as addition or sorting. Their notion of program traces is *different* from our dynamic execution traces consisting of program states with variable valuations. Our notion offers the following advantages: (1) a sequence of program states can be viewed as a sequence of input-output pairs of each executed statement, in other words, sequences of program states provide more robust information than that from sequences of executed statements, and (2) although a sequence of executed statements follows dynamic execution, it is still represented syntactically, and therefore may not adequately capture program semantics. For example, consider the two sorting algorithms in Figure 1. According to Reed & De Freitas (2015), they will have an identical representation *w.r.t.* statements that modify the variable A, *i.e.* a repetition of $A[j] = A[j+1]$ and $A[j+1] = tmp$ for eight times. Our representation, on the other hand, can capture their semantic differences in terms of program states by also only considering the valuation of the variable A.

We have evaluated our dynamic program embeddings in the context of automated program repair. In particular, we use the program embeddings to classify the type of mistakes students made to their programming assignments based on a set of common error patterns (described in the appendix). The dataset for the experiments consists of the programming submissions made to Module 2 assignment in Microsoft-DEV204.1X and two additional problems from the Microsoft CodeHunt platform. The results show that our dynamic embeddings significantly outperform syntax-based program embeddings, including those trained on token sequences and abstract syntax trees. In addition, we show that our dynamic embeddings can be leveraged to significantly improve the efficiency of a search-based program corrector SARFGEN[1] (Wang et al., 2017) (the algorithm is presented in the appendix). More importantly, we believe that our dynamic program embeddings can be useful for many other program analysis tasks, such as program synthesis, fault localization, and similarity detection.

To summarize, the main contributions of this paper are: (1) we show the fundamental limitation of representing programs using syntax-level features; (2) we propose dynamic program embeddings learned from runtime execution traces to overcome key issues with syntactic program representations; (3) we evaluate our dynamic program embeddings for predicting common mistake patterns students make in program assignments, and results show that the dynamic program embeddings outperform state-of-the-art syntactic program embeddings; and (4) we show how the dynamic program embeddings can be utilized to improve an existing production program repair system.

## 2 BACKGROUND: DYNAMIC PROGRAM ANALYSIS

This section briefly reviews dynamic program analysis (Ball, 1999), an influential program analysis technique that lays the foundation for constructing our new program embeddings.

Unlike static analysis (Nielson et al., 1999), *i.e.*, the analysis of program source code, dynamic analysis focuses on program executions. An execution is modeled by a set of atomic actions, or events,

---

[1]Currently integrated with Microsoft-DEV204.1X as a feedback generator for production use.

organized as a trace (or event history). For simplicity, this paper considers sequential executions only (as opposed to parallel executions) which lead to a single sequence of events, specifically, the executions of statements in the program. Detailed information about executions is often not readily available, and separate mechanisms are needed to capture the tracing information. An often adopted approach is to instrument a program's source code (*i.e.*, by adding additional monitoring code) to record the execution of statements of interest. In particular, those inserted instrumentation statements act as a monitoring window through which the values of variables are inspected. This instrumentation process can occur in a fully automated manner, *e.g.*, a common approach is to traverse a program's abstract syntax tree and insert "write" statements right after each program statement that causes a side-effect (*i.e.*, changing the values of some variables).

Consider the two sorting algorithms depicted in Figure 1. If we assume $A$ to be the only variable of interest and subject to monitoring, we can instrument the two algorithms with `Console.WriteLine(A)` after each program location in the code whenever $A$ is modified[2] (*i.e.* the lines marked by comments). Given the input vector $A = [8, 5, 1, 4, 3]$, the execution traces of the two sorting routines are shown on the right in Figure 1.

One of the key benefits of dynamic analysis is its ability to easily and precisely identify relevant parts of the program that affect execution behavior. As shown in the example above, despite the very similar program syntax of bubble sort and insertion sort, dynamic analysis is able to discover their distinct program semantics by exposing their execution traces. Since understanding program semantics is a central issue in program analysis, dynamic analysis has seen remarkable success over the past several decades and has resulted in many successful program analysis tools such as debuggers, profilers, monitors, or explanation generators.

## 3 OVERVIEW OF THE APPROACH

We now present an overview of our approach. Given a program and the execution traces extracted for all its variables, we introduce three neural network models to learn dynamic program embeddings. To demonstrate the utility of these embeddings, we apply them to predict common error patterns (detailed in Section 5) that students make in their submissions to an online introductory programming course.

***Variable Trace Embedding***    As shown in Table 1, each row denotes a new program point where a variable gets updated.[3] The entire variable trace consists of those variable values at all program points. As a subsequent step, we split the complete trace into a list of sub-traces (one for each variable). We use one single RNN to encode each sub-trace independently and then perform max pooling on the final states of the same RNN to obtain the program embedding. Finally, we add a one layer softmax regression to make the predictions. The entire workflow is show in Figure 3.

***State Trace Embedding***    Because each variable trace is handled individually in the previous approach, variable dependencies/interactions are not precisely captured. To address this issue, we propose the state trace embedding. As depicted in Table 1, each program point $l$ introduces a new program state expressed by the latest variable valuations at $l$. The entire state trace is a sequence of program states. To learn the state trace embedding, we first use one RNN to encode each program state (*i.e.*, a tuple of values) and feed the resulting RNN states as a sequence to another RNN. Note that we do not assume that the order in which variables values are encoded by the RNN for each program state but rather maintain a consistent order throughout all program states for a given trace. Finally, we feed a softmax regression layer with the final state of the second RNN (shown in Figure 4). The benefit of state trace embedding is its ability to capture dependencies among variables in each program state as well as the relationship among program states.

***Dependency Enforcement for Variable Trace Embedding***    Although state trace embedding can better capture program dependencies, it also comes with some challenges, the most significant of which is redundancy. Consider a looping structure in a program. During an iteration, whenever

---

[2]On the abstract syntax trees to enable complete automation regardless of the structure of programs.
[3]We ignore the input variable arr since it is read-only (similarly for the state trace later).

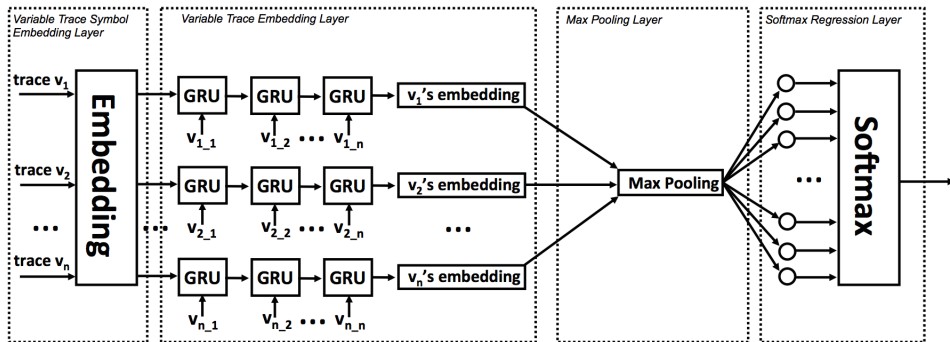

Figure 3: Variable trace for program embedding.

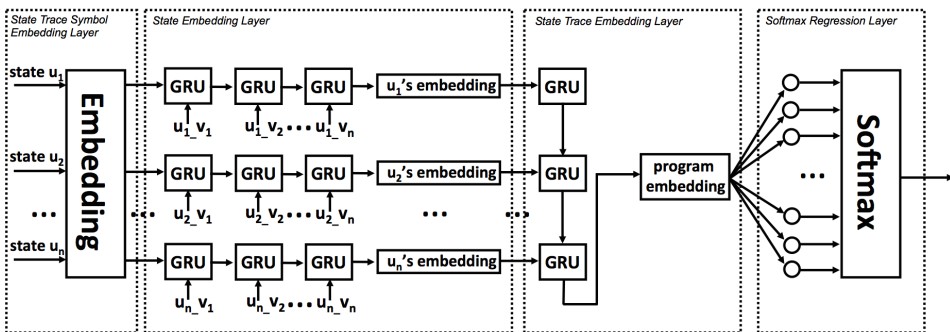

Figure 4: State trace for program embedding.

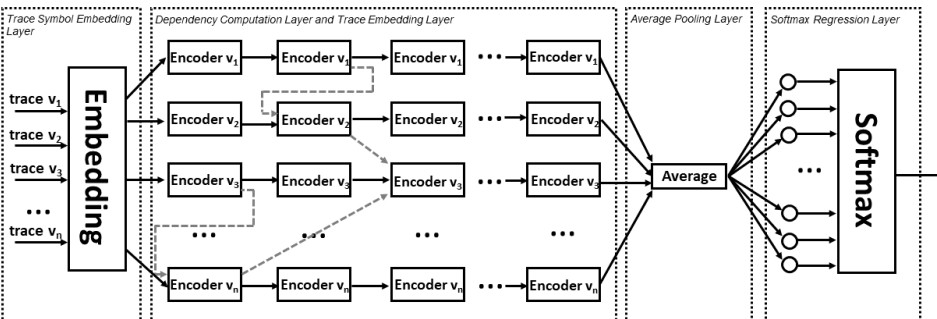

Figure 5: Dependency enforcement embedding. Dotted lines denoted dependencies.

one variable gets modified, a new program state will be created containing the values of all variables, even of those unmodified by the loop. This issue becomes more severe for loops with larger numbers of iterations. To tackle this challenge, we propose the third and final approach, dependency enforcement for variable trace embedding (hereinafter referred as *dependency enforcement embedding*), that combines the advantages of variable trace embedding (*i.e.*, compact representation of execution traces) and state trace embedding (*i.e.*, precise capturing of program dependencies). In dependency enforcement embedding, a program is represented by separate variable traces, with each variable being handled by a different RNN. In order to enforce program dependencies, the hidden states from different RNNs will be interleaved in a way that simulates the needed data and control dependencies. Unlike variable trace embedding, we perform an average pooling on the final states of all RNNs to obtain the program embedding on which we build the final layer of softmax regression. Figure 5 describes the workflow.

## 4 DYNAMIC PROGRAM EMBEDDINGS

We now formally define the three program embedding models.

### 4.1 VARIABLE TRACE MODEL

Given a program $P$, and its variable set $V$ ($v_0$, $v_1$,..., $v_n \in V$), a variable trace is a sequence of values a variable has been assigned during the execution of $P$.[4] Let $x_{t\_v_n}$ denote the value from the variable trace of $v_n$ that is fed to the RNN encoder (Gated Recurrent Unit) at time $t$ as the input, and $h_{t\_v_n}$ as the resulting RNN's hidden state. We compute the variable trace embedding for $P$ in Equation (3) as follows ($h_{T\_v_n}$ denotes the last hidden state of the encoder):

$$h_{t\_v_1} = \text{GRU}(h_{t-1\_v_1}, x_{t\_v_1}) \qquad (1)$$
$$...$$
$$h_{t\_v_n} = \text{GRU}(h_{t-1\_v_n}, x_{t\_v_n}) \qquad (2)$$

$$\text{Evidence} = (\mathbf{W}h_P + b) \qquad (4)$$

$$h_P = \text{MaxPooling}(h_{T\_v_1}, ..., h_{T\_v_n}) \quad (3)$$

$$\text{Y} = \text{softmax}(\text{Evidence}) \qquad (5)$$

We compute the representation of the program trace by performing max pooling over the last hidden state representation of each variable trace embedding. The hidden states $h_{t\_v_1}, \ldots, h_{t\_v_n}, h_P \in \mathbb{R}^k$ where $k$ denotes the size of hidden layers of the RNN encoder. Evidence denotes the output of a linear model through the program embedding vector $h_P$, and we obtain the predicted error pattern class $Y$ by using a softmax operation.

### 4.2 STATE TRACE MODEL

The key idea in state trace model is to embed each program state as a numerical vector first and then feed all program state embeddings as a sequence to another RNN encoder to obtain the program embedding. Suppose $x_{t\_v_n}$ is the value of variable $v_n$ at $t$-th program state, and $h_{t\_v_n}$ is the resulting hidden state of the program state encoder. Equation (8) computes the $t$-th program state embedding. Equations (9-11) encode the sequence of all program state embeddings (*i.e.*, $h_{t\_v_n}$, $h_{t+1\_v_n}$, ..., $h_{t+m\_v_n}$) with another RNN to compute the program embedding.

$$h_{t\_v_1} = \text{GRU}(h_{t\_v_0}, x_{t\_v_1}) \qquad (6)$$

$$h'_{t\_v_n} = \text{GRU}(h'_{t-1\_v_n}, h_{t\_v_n}) \qquad (9)$$

$$h_{t\_v_2} = \text{GRU}(h_{t\_v_1}, x_{t\_v_2}) \qquad (7)$$

$$h'_{t+1\_v_n} = \text{GRU}(h'_{t\_v_n}, h_{t+1\_v_n}) \qquad (10)$$

$$...$$

$$...$$

$$h_{t\_v_n} = \text{GRU}(h_{t\_v_{n-1}}, x_{t\_v_n}) \qquad (8)$$

$$h_P = \text{GRU}(h'_{t+m-1\_v_n}, x_{t+m\_v_n}) \quad (11)$$

$h_{t\_v_1}, \ldots, h_{t\_v_n} \in \mathbb{R}^{k_1}$; $h'_{t\_v_n}, \ldots, h_P \in \mathbb{R}^{k_2}$ where $k_1$ and $k_2$ denote, respectively, the sizes of hidden layers of the first and second RNN encoders.

### 4.3 DEPENDENCY ENFORCEMENT FOR VARIABLE TRACE EMBEDDING

The motivation behind this model is to combine the advantages of the previous two approaches, *i.e.* representing the execution trace compactly while enforcing the dependency relationship among variables as much as possible. In this model, each variable trace is handled with a different RNN. A potential issue to be addressed is variable matching/renaming (*i.e.*, $\alpha$-renaming). In other words same variables may be named differently in different programs. Processing each variable id with a single RNN among all programs in the dataset will not only cause memory issues, but more importantly the loss of precision. Our solution is to (1) execute all programs to collect traces for all variables, (2) perform *dynamic time wrapping* (Vintsyuk, 1968) on the variable traces across all programs to find the top-$n$ most used variables that account for the vast majority of variable usage, and (3) rename the top-$n$ most used variables consistently across all programs, and rename all other variables to a same special variable.

---

[4]For presentation simplicity and w.l.o.g., we assume that the program does not take any inputs.

Given the same set of variables among all programs, the mechanism of dependency enforcement on the top ones is to fuse the hidden states of multiple RNNs based on how a new value of a variable is produced. For example, in Figure 2 at line 8, the new value of $max\_val$ is data-dependent on $item$, and control-dependent on both $item$ and itself. So at the time step when the new value of $max\_val$ is produced, the latest hidden states of the RNNs encode variable $item$ as well as itself; they together determine the previous state of the RNN upon which the new value of $max\_val$ is produced. If a value is produced without any dependencies, this mechanism will not take effect. In other words, the RNN will act normally to handle data sequences on its own. In this work we enforce the data-dependency in assignment statement, declaration statement and method calls; and control-dependency in control statements such as $if$, $for$ and $while$ statements. Equations (11 and 12) expose the inner workflow. $h_{LT\_v_m}$ denotes the latest hidden state of the RNN encoding variable trace of $v_m$ up to the point of time $t$ when $x_{t\_v_n}$ is the input of the RNN encoding variable trace of $v_n$. $\odot$ denotes element-wise matrix product.

$$h_{t-1\_v_n} = h_{LT\_v_1} \odot h_{LT\_v_m} \odot h_{LT\_v_n} \qquad \text{Given } v_n \text{ depends on } v_1 \text{ and } v_m \qquad (11)$$
$$h_{t\_v_n} = \text{GRU}(h_{t-1\_v_n}, x_{t\_v_n}) \qquad (12) \qquad h_P = \text{AveragePooling}(h_{T\_v_1}, ..., h_{T\_v_n}) \qquad (13)$$

## 5 EVALUATION

We train our dynamic program embeddings on the programming submissions obtained from Assignment 2 from Microsoft-DEV204.1X: "Introduction to C#" offered on edx and two other problems on Microsoft CodeHunt platform.

- **Print Chessboard**: Print the chessboard pattern using "X" and "O" to represent the squares as shown in Figure 6.

- **Count Parentheses**: Count the depth of nesting parentheses in a given string.

- **Generate Binary Digits**: Generate the string of binary digits for a given integer.

```
XOXOXOXO
OXOXOXOX
XOXOXOXO
OXOXOXOX
XOXOXOXO
OXOXOXOX
XOXOXOXO
OXOXOXOX
```

Figure 6: The desired output for the chessboard exercise.

Regarding the three programming problems, the errors students made in their submissions can be roughly classified into low-level technical issues (*e.g.*, list indexing, branching conditions or looping bounds) and high-level conceptual issues (*e.g.*, mishandling corner case, misunderstanding problem requirement or misconceptions on the underlying data structure of test inputs).[5]

In order to have sufficient data for training our models to predict the error patterns, we (1) convert each incorrect program into multiple programs such that each new program will have only one error, and (2) mutate all the correct programs to generate synthetic incorrect programs such that they exhibit similar errors that students made in real program submissions. These two steps allow us to set up a dataset depicted in Table 2. Based on the same set of training data, we evaluate the dynamic embeddings trained with the three network models and compare them with the syntax-based program embeddings (on the same error prediction task) on the same testing data. The syntax-based models include (1) one trained with a RNN that encodes the run-time syntactic traces of programs (Reed & De Freitas, 2015); (2) another trained with a RNN that encodes token sequences of programs; and (3) the third trained with a RNN on abstract syntax trees of programs (Socher et al., 2013).

| Problem | Program Submissions | | Synthetic Data | | |
|---|---|---|---|---|---|
| | Correct | Incorrect | Training | Validation | Testing |
| Print Chessboard | 2,281 | 742 | 120K | 13K | 15K |
| Count Parentheses | 505 | 315 | 20K | 2K | 2K |
| Generate Binary Digits | 518 | 371 | 22K | 3K | 2K |

Table 2: Dataset for experimental evaluation.

---

[5]Please refer to the Appendix for a detailed summary of the error patterns for each problem.

All models are implemented in TensorFlow. All encoders in each of the trace model have two stacked GRU layers with 200 hidden units in each layer except that the state encoder in the state trace model has one single layer of 100 hidden units. We adopt random initialization for weight initialization. Our vocabulary has 5,568 unique tokens (*i.e.*, the values of all variables at each time step), each of which is embedded into a 100-dimensional vector. All networks are trained using the Adam optimizer (Kingma & Ba, 2014) with the learning and the decay rates set to their default values (learning_rate = 0.0001, beta1 = 0.9, beta2 = 0.999) and a mini-batch size of 500. For the variable trace and dependency enforcement models, each trace is padded to have the same length across each batch; for the state trace model, both the number of variables in each program state as well as the length of the entire state trace are padded.

During the training of the dependency enforcement model, we have observed that when dependencies become complex, the network suffers from optimization issues, such as diminishing and exploding gradients. This is likely due to the complex nature of fusing hidden states among RNNs, echoing the errors back and forth through the network. We resolve this issue by truncating each trace into multiple sub-sequences and only back-propagate on the last sub-sequence while only feed-forwarding on the rest. Regarding the baseline network trained on syntactic traces/token sequences, we use the same encoder architecture (*i.e.*, two layer GRU of 200 hidden units) processing the same 100-dimension embedding vector for each statement/token. As for the AST model, we learn an embedding (100-dimension) for each type of the syntax node by propagating the leaf (a simple look up) to the root through the learned production rules. Finally, we use the root embeddings to represent programs.

| Programming Problem | Variable Trace | State Trace | Dependency Enforcement | Run-Time Syntactic Trace | Token | AST |
|---|---|---|---|---|---|---|
| Print Chessboard | 93.9% | 95.3% | 99.3% | 26.3% | 16.8% | 16.2% |
| Count Parentheses | 92.7% | 93.8% | 98.8% | 25.5% | 19.3% | 21.7% |
| Generate Binary Digits | 92.1% | 94.5% | 99.2% | 23.8% | 21.2% | 20.9% |

Table 3: Comparing dynamic program embeddings with syntax-based program embedding in predicting common error patterns made by students.

As shown in Table 3, our embeddings trained on execution traces significantly outperform those trained on program syntax (greater than $92\%$ accuracy compared to less than $27\%$ for syntax-based embeddings). We conjecture this is because of the fact that minor syntactic discrepancies can lead to major semantic differences as shown in Figure 1. In our dataset, there are a large number of programs with distinct labels that differ by only a few number of tokens or AST nodes, which causes difficulty for the syntax models to generalize. Even for the simpler syntax-level errors, they are buried in large number of other syntactic variations and the size of the training dataset is relatively small for the syntax-based models to learn precise patterns. In contrast, dynamic embeddings are able to canonicalize the syntactical variations and pinpoint the underlying semantic differences, which results in the trace-based models learning the correct error patterns more effectively even with relatively smaller size of the training data.

In addition, we incorporated our dynamic program embeddings into SARFGEN (Wang et al., 2017) — a program repair system — to demonstrate their benefit in producing fixes to correct students errors in programming assignments. Given a set of potential repair candidates, SARFGEN uses an enumerative search-based technique to find minimal changes to an incorrect program. We use the dynamic embeddings to learn a distribution over the corrections to prioritize the search for the repair algorithm.[6] To establish the baseline, we obtain the set of all corrections from SARFGEN for each of the real incorrect program to all three problems and enumerate each subset until we find the minimum fixes. On the contrary, we also run another experiment where we prioritize each correction according to the prediction of errors with the dynamic embeddings. It is worth mentioning that one incorrect program may be caused by multiple errors. Therefore, we only predict the top-1 error each time and repair the program with the corresponding corrections. If the program is still incorrect, we repeat this procedure till the program is fixed. The comparison between the two approaches is based on how long it takes them to repair the programs.

---

[6] Some corrections are merely syntactic discrepancies (*i.e.*, they do not change program semantics such as modifying $a *= 2$ to $a += a$). In order to provide precise fixes, those false positives would need to be eliminated.

| Number of Fixes | Enumerative Search | Variable Trace Embeddings | State Trace Embeddings | Dependency Enforcement Embeddings |
|---|---|---|---|---|
| 1-2 | 3.8 | 2.5 | 2.8 | 3.3 |
| 3-5 | 44.7 | 3.6 | 3.1 | 4.1 |
| 6-7 | 95.9 | 4.2 | 3.6 | 4.5 |
| ≥8 | 128.3 | 41.6 | 49.5 | 38.8 |

Table 4: Comparing the enumerative search with those guided by dynamic program embeddings in finding the minimum fixes. Time is measured in seconds.

As shown in Table 4, the more fixes required, the more speedups dynamic program embeddings yield — more than an order of magnitude speedups when the number of fixes is four or greater. When the number of fixes is greater than seven, the performance gain drops significantly due to poor prediction accuracy for programs with too many errors. In other words, our dynamic embeddings are not viewed by the network as capturing incorrect execution traces, but rather new execution traces. Therefore, the predictions become unreliable. Note that we ignored incorrect programs having greater than 10 errors when most experiments run out of memory for the baseline approach.

## 6 RELATED WORK

There has been significant recent interest in learning neural program representations for various applications, such as program induction and synthesis, program repair, and program completion. Specifically for neural program repair techniques, none of the existing techniques, such as Deep-Fix (Gupta et al., 2017), SynFix (Bhatia & Singh, 2016) and sk_p (Pu et al., 2016), have considered dynamic embeddings proposed in this paper. In fact, dynamic embeddings can be naturally extended to be a new feature dimension for these existing neural program repair techniques.

Piech et al. (2015) is a notable recent effort targeting program representation. Piech *et al.* explore the possibility of using input-output pairs to represent a program. Despite their new perspective, the direct mapping between input and output of programs usually are not precise enough, *i.e.*, the same input-output pair may correspond to two completely different programs, such as the two sorting algorithms in Figure 1. As we often observe in our own dataset, programs with the same error patterns can also result in different input-output pairs. Their approach is clearly ineffective for these scenarios.

Reed & De Freitas (2015) introduced the novel approach of using execution traces to induce and execute algorithms, such as addition and sorting, from very few examples. The differences from our work are (1) they use a sequence of instructions to represent dynamic execution trace as opposed to using dynamic program states; (2) their goal is to synthesize a neural controller to execute a program as a sequence of actions rather than learning a semantic program representation; and (3) they deal with programs in a language with low-level primitives such as function stack push/pop actions rather than a high-level programming language.

As for learning representations, there are several related efforts in modeling semantics in sentence or symbolic expressions (Socher et al., 2013; Zaremba et al., 2014; Bowman, 2013). These approaches are similar to our work in spirit, but target different domains than programs.

## 7 CONCLUSION

We have presented a new program embedding that learns program representations from runtime execution traces. We have used the new embeddings to predict error patterns that students make in their online programming submissions. Our evaluation shows that the dynamic program embeddings significantly outperform those learned via program syntax. We also demonstrate, via an additional application, that our dynamic program embeddings yield more than 10x speedups compared to an enumerative baseline for search-based program repair. Beyond neural program repair, we believe that our dynamic program embeddings can be fruitfully utilized for many other neural program analysis tasks such as program induction and synthesis.

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

APPENDIX

ERROR PATTERNS

Print Chessboard:

- Misprinting "O" to "0" or printing lower case instead of upper case characters.
- Switching across rows are supposed to be the other way around ( *i.e.* printing `OXOXOXOX` for odd number rows and `XOXOXOXO` for even number rows).
- Printing the first row correctly but failed to make a switch across rows.
- Printing the entire chessboard as "X" or "O" only.
- Printing the chessboard correctly but with extra unnecessary characters.
- Printing the incorrect number of rows.
- Printing the incorrect number of columns.
- Printing the characters correctly but in wrong format (*i.e.* not correctly seperated with the spaces to form the rows).
- Others.

Count Parentheses:

- Miss the corner case of empty strings.
- Mistakenly consider the parenthesis to be symbols rather than "(" or ")".
- Mishandling the string of unmatched parentheses.
- Counting the number of matching parentheses rather then depth.
- Incorrectly assume nested parentheses are always present.
- Miscounting the characters which should have been ignored.
- Others.

Generate Binary Digits:

- Miss the corner case of integer 0.
- Misunderstand the binary digits to be underlying bytes of a string.
- Mistakes in arithmetic calculation regrading shift operations.
- Adding the binary digits rather than concatenating them to a string.
- Miss the one on the most significant bit.
- Others.

SARFGEN'S ALGORITHM

---

**Algorithm 1:** SARFGEN 's feedback generation procedure.

```
/* P:  an incorrect program; P_s:  all correct solutions        */
```
1  **function** FixGeneration(*P, P_s*)
2     **begin**
   ```
   // Among P_s identify P_cs to be reference programs to fix P
   ```
3        $P_{cs} \leftarrow$ CandidatesIdentification($P, P_s$)
   ```
   // Initialize the minimum number of fixes k to be inifinity
   ```
4        $k \leftarrow \infty$
   ```
   // Initialize the minimum set of fixes F(P)
   ```
5        $\mathcal{F}(P) \leftarrow$ null
6        **for** $P_c \in P_{cs}$ **do**
   ```
   // Generates the syntactic discrepencies w.r.t. each P_c
   ```
7           $\mathcal{C}(P, P_c) \leftarrow$ DiscrepenciesGeneration($P, P_s$)
   ```
   // Selecting subsets of C(P, P_c) from size of one itll |C(P, P_c)|
   ```
8           **for** $n \in [1, 2, ..., |\mathcal{C}(P, P_c)|]$ **do**
9              $\mathcal{C}_{subs}(P, P_c) \leftarrow \{x \mid x \subseteq \mathcal{C}(P, P_c) \wedge |x| = n\}$
   ```
   // Attemp each subset of C(P, P_c)
   ```
10             **for** $\mathcal{C}_{sub}(P, P_c) \in \mathcal{C}_{subs}(P, P_c)$ **do**
11                $P' \leftarrow$ PatchApplication($P, \mathcal{C}_{sub}(P, P_c)$)
   ```
   // Update k if necessary
   ```
12                **if** *isCorrect(P')* **then**
13                   **if** $|P'| < k$ **then**
14                      $k \leftarrow |P'|$
15                      $\mathcal{F}(P) \leftarrow P'$

16       **return** $\mathcal{F}(P)$

---

**Algorithm 2:** Incorporate pre-trained model to SARFGEN 's feedback generation procedure.

```
/* P, P_s:  same as above; M:  learned Model                    */
```
1  **function** FixGeneration(*P, P_s, $\mathcal{M}$*)
2     **begin**
   ```
   // Among P_s identify P_cs to be reference programs to fix P
   ```
3        $P_{cs} \leftarrow$ CandidatesIdentification($P, P_s$)
   ```
   // Initialize the minimum number of fixes k to be inifinity
   ```
4        $k \leftarrow \infty$
   ```
   // Initialize the minimum set of fixes F(P)
   ```
5        $\mathcal{F}(P) \leftarrow$ null
6        **for** $P_c \in P_{cs}$ **do**
   ```
   // Generates the syntactic discrepencies w.r.t. each P_c
   ```
7           $\mathcal{C}(P, P_c) \leftarrow$ DiscrepenciesGeneration($P, P_s$)
   ```
   // Executing P to extract the dynamic execution trace
   ```
8           $\mathcal{T}(P) \leftarrow$ DynamicTraceExtraction($P$)
   ```
   // Prioritizing subsets of C(P, P_c) through pre-trained model
   ```
9           $\mathcal{C}_{subs}(P, P_c) \leftarrow$ Prioritization($\mathcal{C}(P, P_c), \mathcal{T}(P), \mathcal{M}$)
10          **for** $\mathcal{C}_{sub}(P, P_c) \in \mathcal{C}_{subs}(P, P_c)$ **do**
11             $P' \leftarrow$ PatchApplication($P, \mathcal{C}_{sub}(P, P_c)$)
12             **if** *isCorrect(P')* **then**
13                **if** $|P'| < k$ **then**
14                   $k \leftarrow |P'|$
15                $\mathcal{F}(P) \leftarrow P'$

16       **return** $\mathcal{F}(P)$

---

