# OpenReview forum: "Dynamic Neural Program Embeddings for Program Repair"
_ICLR.cc/2018/Conference — Accept (Poster)_

### Official Review · AnonReviewer1 · 2017-11-28
**A solid paper with some clarity issues**

**Rating:** 7
**Confidence:** 4

**Review:**

The authors present 3 architectures for learning representations of programs from execution traces. In the variable trace embedding, the input to the model is given by a sequence of variable values. The state trace embedding combines embeddings for variable traces using a second recurrent encoder. The dependency enforcement embedding performs element-wise multiplication of embeddings for parent variables to compute the input of the GRU to compute the new hidden state of a variable. The authors evaluate their architectures on the task of predicting error patterns for programming assignments from Microsoft DEV204.1X (an introduction to C# offered on edx) and problems on the Microsoft CodeHunt platform. They additionally use their embeddings to decrease the search time for the Sarfgen program repair system.

This is a fairly strong paper. The proposed models make sense and the writing is for the most part clear, though there are a few places where ambiguity arises:

- The variable "Evidence" in equation (4) is never defined.

- The authors refer to "predicting the error patterns", but again don't define what an error pattern is. The appendix seems to suggest that the authors are simply performing multilabel classification based on a predefined set of classes of errors, is this correct?

- It is not immediately clear from Figures 3 and 4 that the architectures employed are in fact recurrent.

- Figure 5 seems to suggest that dependencies are only enforced at points in a program where assignment is performed for a variable, is this correct?

Assuming that the authors can address these clarity issues, I would in principle be happy for the paper to appear.

---

> ### Author Response · Authors · 2017-12-12
> **Response to AnonReviewer1**
>
> Thank you for the helpful suggestions.  Below, we answer the questions that you raised in the review.
>
> Our revision will clarify the definition of the “Evidence” variable, which, in short, denotes the result of multiplying weight on the program embedding vector and then adding the bias.
>
> Yes, “predicting the error patterns” means classifying the kinds of errors that students made in their programs.
>
> The encoders in Figures 3 and 4 are recurrent as they encode variable traces (each variable trace is a sequence of variable values) and states (a state is a set of variable values at a particular program location). The figures in our revision will make these clearer.
>
> Dependencies happen primarily in assignment statements. API calls with side effects also introduce dependencies. For example, in the code snippet below,  “sb” depends on “s”:
>
> StringBuilder sb = new StringBuilder();
> String s = “str”;
> sb.Append(s);

---

> ### Author Response · Authors · 2017-12-23
> **Response**
>
> Dear reviewer:
>
> We have uploaded a revision of our paper that incorporates (1) the requested clarifications in the reviews and (2) additional experimental results from comparing our embeddings with syntactic trace based program embeddings (Reed & De Freitas (2015) and Cai et. al (2017)).  Please let us know if any further clarifications are needed.

---

### Official Review · AnonReviewer3 · 2017-11-28
**.**

**Rating:** 7
**Confidence:** 3

**Review:**

Summary of paper: The paper proposes an RNN-based neural network architecture for embedding programs, focusing on the semantics of the program rather than the syntax. The application is to predict errors made by students on programming tasks. This is achieved by creating training data based on program traces obtained by instrumenting the program by adding print statements. The neural network is trained using this program traces with an objective for classifying the student error pattern (e.g. list indexing, branching conditions, looping bounds).

---

Quality: The experiments compare the three proposed neural network architectures with two syntax-based architectures. It would be good to see a comparison with some techniques from Reed & De Freitas (2015) as this work also focuses on semantics-based embeddings.
Clarity: The paper is clearly written.
Originality: This work doesn't seem that original from an algorithmic point of view since Reed & De Freitas (2015) and Cai et. al (2017) among others have considered using execution traces. However the application to program repair is novel (as far as I know).
Significance: This work can be very useful for an educational platform though a limitation is the need for adding instrumentation print statements by hand.

---

Some questions/comments:
- Do we need to add the print statements for any new programs that the students submit? What if the structure of the submitted program doesn't match the structure of the intended solution and hence adding print statements cannot be automated?

---

References

Cai, J., Shin, R., & Song, D. (2017). Making Neural Programming Architectures Generalize via Recursion. In International Conference on Learning Representations (ICLR).

---

> ### Author Response · Authors · 2017-12-12
> **Response to AnonReviewer3**
>
> We appreciate your point on the differences between our work and Reed & De Freitas (2015).  We have given a detailed discussion regarding this point in our response to AnonReviewer2, which we include below for your convenience.
>
> There are fundamental differences between the syntactic program traces explored in prior work (Reed & De Freitas (2015)) and the “semantic program traces” considered in our work. Consider the example in Figure 1. According to Reed & De Freitas (2015), the two sorting algorithms will have an identical representation with respect to statements that modify the variable A:
>
> A[j] = A[j + 1]
> A[j + 1] = tmp
> A[j] = A[j + 1]
> A[j + 1] = tmp
> A[j] = A[j + 1]
> A[j + 1] = tmp
> A[j] = A[j + 1]
> A[j + 1] = tmp
> A[j] = A[j + 1]
> A[j + 1] = tmp
> A[j] = A[j + 1]
> A[j + 1] = tmp
> A[j] = A[j + 1]
> A[j + 1] = tmp
> A[j] = A[j + 1]
> A[j + 1] = tmp
>
> Our representation, on the other hand, can capture their semantic differences in terms of program states by also only considering the variable A:
>
>   Bubble          Insertion
> [5,5,1,4,3]	[5,5,1,4,3]
> [5,8,1,4,3]	[5,8,1,4,3]
> [5,1,1,4,3] 	[5,1,1,4,3]
> [5,1,8,4,3] 	[5,1,8,4,3]
> [1,1,8,4,3] 	[5,1,4,4,3]
> [1,5,8,4,3] 	[5,1,4,8,3]
> [1,5,4,4,3]	[5,1,4,3,3]
> [1,5,4,8,3] 	[5,1,4,3,8]
> [1,4,4,8,3] 	[1,1,4,3,8]
> [1,4,5,8,3] 	[1,5,4,3,8]
> [1,4,5,3,3] 	[1,4,4,3,8]
> [1,4,5,3,8] 	[1,4,5,3,8]
> [1,4,3,3,8] 	[1,4,3,3,8]
> [1,4,3,5,8] 	[1,4,3,5,8]
> [1,3,3,5,8] 	[1,3,3,5,8]
> [1,3,4,5,8] 	[1,3,4,5,8]
>
> This example also illustrates concretely the point made in Section 1 that minor syntactic differences can lead to signiﬁcant semantic differences. Therefore, the approach of Reed & De Freitas is insufficient to capture such semantic differences.  As another example, consider the following two programs:
>
> static void Main(string[] args)
> {
>         string str = String.Empty;
>         int x = 0;
>         x++;
> }
>
> static void Main(string[] args)
> {
>         string s = "";
>         int y = 0;
>         y = y+1;
> }
>
> According to the representation proposed in Reed & De Freitas (2015), the first program is represented as [string str = String.Empty, int x = 0, x++], while the second represented as [string s = "", int y = 0, y = y+1].  Although the two programs share the same semantics, they are represented differently due to syntactic variations. In contrast, our work captures the same semantic trace for both programs, i.e., [ [“”, NA], [“”,0], [“”,1]].
>
> To sum up, the embedding proposed in Reed & De Freitas (2015) is a syntactic representation, and cannot precisely capture a program’s semantics and abstract away its syntactic redundancies. Consequently, the encoder will not be able to learn the true feature dimensions in the embeddings.  We also performed additional experiments to contrast the two trace-based approaches. We used the same configuration of encoder (cf. Section 5) to embed the syntactic traces on the same datasets for the same classification problem. The results are as follows:
>
> Problems                        Reed & De Freitas (2015)          Token                 AST            Dependency Model
> Print Chessboard                   26.3%                                   16.8%                16.2%                    99.3%
> Count Parentheses                25.5%                                   19.3%                21.7%	                 98.8%
> Generate Binary Digits         23.8%                                    21.2%                20.9%	                 99.2%
>
> Although syntactic traces result in better accuracy than Token and AST, they are still significantly worse than semantic embeddings introduced in our work.
>
> Our revision will include the representation proposed in Reed & De Freitas (2015) for the example programs in Figure 1. It will also include the experimental setup (in Section 5) and the new results (in a new column of Table 3).
>
> We will also add a citation to Cai et al. (2017), which uses the exact same program representation as Reed & De Freitas (2015). The other contributions in Cai et al. (2017) are unrelated to our work.
>
> We hope that our response helped address your concerns. Please let us know if you have any additional questions. Thank you.

---

> ### Author Response · Authors · 2017-12-23
> **Response**
>
> Dear reviewer:
>
> Our earlier reply mistakenly omitted our answer to the question in your review.  Our apologies, and we include the answer below. The instrumentation for adding print statements to a program is fully automated, and requires no manual effort or any assumption on the program’s code structure. It traverses the program’s abstract syntax tree and inserts the appropriate print statement after each side-effecting program statement, i.e., a statement that changes the values of some program variables.
>
> Can you please inform us whether there are any additional clarifications needed beyond those in our response?  We have also uploaded a revision of our paper that incorporates (1) the requested clarifications in the reviews and (2) additional experimental results from comparing our embeddings with syntactic trace based program embeddings (Reed & De Freitas (2015) and Cai et. al (2017)).

---

> > ### Comment · AnonReviewer3 · 2018-01-16
> > **Re: Response**
> >
> > Thank you for the clarifications regarding the difference between semantic and syntactic program traces and the extra experiment. I'm bumping up my score to a 7.

---

### Official Review · AnonReviewer2 · 2017-12-02
**Interesting application, but lacks clarity**

**Rating:** 6
**Confidence:** 2

**Review:**

This paper considers the task of learning program embeddings with neural networks with the ultimate goal of bug detection program repair in the context of students learning to program. Three NN architectures are explored, which leverage program semantics rather than pure syntax.  The approach is validated using programming assignments from an online course, and compared against syntax based approaches as a baseline.

The problem considered by the paper is interesting, though it's not clear from the paper that the approach is a substantial improvement over previous work. This is in part due to the fact that the paper is relatively short, and would benefit from more detail.  I noticed the following issues:

1) The learning task is based on error patterns, but it's not clear to me what exactly that means from a software development standpoint.
2) Terms used in the paper are not defined/explained. For example, I assume GRU is gated recurrent unit, but this isn't stated.
3) Treatment of related work is lacking.  For example, the Cai et al. paper from ICLR 2017 is not considered
4) If I understand dependency reinforcement embedding correctly, a RNN is trained for every trace. If so, is this scalable?

I believe the work is very promising, but this manuscript should be improved prior to publication.

---

> ### Author Response · Authors · 2017-12-12
> **Response to AnonReviewer2**
>
> Thank you for the review.  We clarify below the four specific points raised.
>
> 1. By “error patterns”, we mean different types of errors that students made in their programming submissions. This work focuses on providing quality feedback to students.  It may be extended in future work to help software developers, where error patterns can correspond to different classes of errors that developers may make. However, it is not the consideration for the current version of the paper.
>
> 2. We will clarify all abbreviations and terms used in the paper.
>
> Yes, GRU is Gated Recurrent Unit.
>
> 3. The results of our latest experiments clearly indicate that this work substantially improves prior work. We briefly highlight the main reasons below.  First, there are fundamental differences between the syntactic program traces explored in prior work (Reed & De Freitas (2015)) and the “semantic program traces” considered in our work. Consider the example in Figure 1. According to Reed & De Freitas (2015), the two sorting algorithms will have an identical representation with respect to statements that modify the variable A:
>
> A[j] = A[j + 1]
> A[j + 1] = tmp
> A[j] = A[j + 1]
> A[j + 1] = tmp
> A[j] = A[j + 1]
> A[j + 1] = tmp
> A[j] = A[j + 1]
> A[j + 1] = tmp
> A[j] = A[j + 1]
> A[j + 1] = tmp
> A[j] = A[j + 1]
> A[j + 1] = tmp
> A[j] = A[j + 1]
> A[j + 1] = tmp
> A[j] = A[j + 1]
> A[j + 1] = tmp
>
> Our representation, on the other hand, can capture their semantic differences in terms of program states by also only considering the variable A:
>
>  Bubble           Insertion
> [5,5,1,4,3]	[5,5,1,4,3]
> [5,8,1,4,3]	[5,8,1,4,3]
> [5,1,1,4,3] 	[5,1,1,4,3]
> [5,1,8,4,3] 	[5,1,8,4,3]
> [1,1,8,4,3] 	[5,1,4,4,3]
> [1,5,8,4,3] 	[5,1,4,8,3]
> [1,5,4,4,3]	[5,1,4,3,3]
> [1,5,4,8,3] 	[5,1,4,3,8]
> [1,4,4,8,3] 	[1,1,4,3,8]
> [1,4,5,8,3] 	[1,5,4,3,8]
> [1,4,5,3,3] 	[1,4,4,3,8]
> [1,4,5,3,8] 	[1,4,5,3,8]
> [1,4,3,3,8] 	[1,4,3,3,8]
> [1,4,3,5,8] 	[1,4,3,5,8]
> [1,3,3,5,8] 	[1,3,3,5,8]
> [1,3,4,5,8] 	[1,3,4,5,8]
>
> This example also illustrates concretely the point made in Section 1 that minor syntactic differences can lead to signiﬁcant semantic differences. Therefore, the approach of Reed & De Freitas is insufficient to capture such semantic differences.  As another example, consider the following two programs:
>
> static void Main(string[] args)
> {
>         string str = String.Empty;
>         int x = 0;
>         x++;
> }
>
> static void Main(string[] args)
> {
>         string s = "";
>         int y = 0;
>         y = y+1;
> }
>
> According to the representation proposed in Reed & De Freitas (2015), the first program is represented as [string str = String.Empty, int x = 0, x++], while the second represented as [string s = "", int y = 0, y = y+1].  Although the two programs share the same semantics, they are represented differently due to syntactic variations. In contrast, our work captures the same semantic trace for both programs, i.e., [ [“”, NA], [“”,0], [“”,1]].
>
> To sum up, the embedding proposed in Reed & De Freitas (2015) is a syntactic representation, and cannot precisely capture a program’s semantics and abstract away its syntactic redundancies. Consequently, the encoder will not be able to learn the true feature dimensions in the embeddings.  We also performed additional experiments to contrast the two trace-based approaches. We used the same configuration of encoder (cf. Section 5) to embed the syntactic traces on the same datasets for the same classification problem. The results are as follows:
>
> Problems                        Reed & De Freitas (2015)          Token                 AST            Dependency Model
> Print Chessboard                   26.3%                                   16.8%                16.2%                    99.3%
> Count Parentheses                25.5%                                   19.3%                21.7%	                 98.8%
> Generate Binary Digits          23.8%                                   21.2%               20.9%	                 99.2%
>
> Although syntactic traces result in better accuracy than Token and AST, they are still significantly worse than semantic embeddings introduced in our work.
>
> Our revision will include the representation proposed in Reed & De Freitas (2015) for the example programs in Figure 1. It will also include the experimental setup (in Section 5) and the new results (in a new column of Table 3).
>
> We will also add a citation to Cai et al. (2017), which uses the exact same program representation as Reed & De Freitas (2015). The other contributions in Cai et al. (2017) are unrelated to our work.
>
> 4. The first paragraph of Section 4.3 addresses the scalability of the dependency architecture that you questioned.
> “ ...Processing each variable id with a single RNN among all programs in the dataset will not only cause memory issues, but more importantly the loss of precision…”
>
> We hope that our response helped address your concerns. Please let us know if you have any additional questions. Thank you.

---

> ### Author Response · Authors · 2017-12-23
> **Response**
>
> Dear reviewer:
>
> Can you please inform us whether there are any additional clarifications needed beyond those in our response?  We have also uploaded a revision of our paper that incorporates (1) the requested clarifications in the reviews and (2) additional experimental results from comparing our embeddings with syntactic trace based program embeddings (Reed & De Freitas (2015) and Cai et. al (2017)).

---

### Author Response · Authors · 2017-12-26
**Revision Summary**

We thank the reviewers for their helpful comments and feedback, and suggestions for additional experiments. We have uploaded a new revision of the paper with the following revisions:

1. We incorporated the requested clarification questions/definitions in the reviews including defining terms/variables, new figures to show the recurrent models for variable and state trace embeddings, defining error patterns and formulating the repair problem as one of classification, automated instrumentation of programs, data-dependency in traces, training details etc.

2. We added more descriptive examples to showcase the difference between the "semantic program traces" (in terms of variable valuations) considered in this work compared to previous works (Reed & De Freitas (2015) and Cai et al. (2017)) that consider "syntactic traces".

3. We added additional experimental results to compare syntactic program trace based embeddings (Reed & De Freitas (2015) and Cai et al. (2017)) in Section 5 (Table 3). Although syntactic traces result in better accuracy than Token and AST (~26% vs ~20%), they are still significantly worse than semantic trace embeddings introduced in our work.

---

### Decision · Program_Chairs · 2018-01-29
**ICLR 2018 Conference Acceptance Decision**

**Decision:**

Accept (Poster)

**Comment:**

PROS:

1. Interesting and clearly useful idea
2. The paper is clearly written.
3. This work doesn't seem that original from an algorithmic point of view since Reed & De Freitas (2015) and Cai et. al (2017) among others have considered using execution traces. However the application to program repair is novel (as far as I know).
4. This work can be very useful for an educational platform though a limitation is the need for adding instrumentation print statements by hand.

CONS:

1. The paper has some clarity issues which the authors have promised to fix.

---